# Ceramization Mechanism of Ceramizable Silicone Rubber Composites with Nano Silica at Low Temperature

**DOI:** 10.3390/ma13173708

**Published:** 2020-08-21

**Authors:** Penghu Li, Haiyun Jin, Shichao Wei, Huaidong Liu, Naikui Gao, Zhongqi Shi

**Affiliations:** 1State Key Laboratory of Electrical Insulation and Power Equipment, Xi’an Jiaotong University, Xi’an 710049, China; li931230@stu.xjtu.edu.cn (P.L.); shichaow@stu.xjtu.edu.cn (S.W.); lhd770322@stu.xjtu.edu.cn (H.L.); gnk@mail.xjtu.edu.cn (N.G.); 2State Key Laboratory for Mechanical Behavior of Materials, Xi’an Jiaotong University, Xi’an 710049, China; zhongqishi@mail.xjtu.edu.cn

**Keywords:** ceramizable composites, polymer matrix composites, nano silica, liquid-phase sintering, microstructure

## Abstract

Ceramizable composite is a kind of polymer matrix composite that can turn into ceramic material at a high temperature. It can be used for the ceramic insulation of a metal conductor because of its processability. However, poor low-temperature ceramization performance is a problem of ceramizable composites. In this paper, ceramizable composites were prepared by using silicone rubber as a matrix. Ceramic samples were sintered at different temperatures no more than 1000 °C, according to thermogravimetric analysis results of the composites. The linear contraction and flexural strength of the ceramics were measured. The microstructure and crystalline phase of ceramics were analyzed using scanning electron microscope (SEM) and X-ray diffraction (XRD). The results show that the composites turned into ceramics at 800 °C, and a new crystal and continuous microstructure formed in the samples. The flexural strength of ceramics was 46.76 MPa, which was more than twice that of similar materials reported in other research sintered at 1000 °C. The maximum flexural strength was 54.56 MPa, when the sintering temperature was no more than 1000 °C. Moreover, glass frit and nano silica played important roles in the formation of the ceramic phase in this research. A proper content of nano silica could increase the strength of the ceramic samples.

## 1. Introduction

A ceramizable silicone rubber composite is a new polymer matrix composite made of silicone rubber, inorganic filler, fluxing agent, reinforcing agent, and so on. The composite has the characteristics of silicone rubber and a good processability at room temperature, and can turn into ceramic with mechanical strength at a high temperature [1,2,3,4,5]. Nowadays, this material is widely used in fire-resistant cables. The composite will turn into ceramic at a high temperature, prevent the spread of flames, and keep cables working normally [6,7,8,9,10]. Because of its good processibility before sintering and lower sintering temperature than traditional ceramics, this material can also be used for the ceramic insulation of a metal conductor such as a bus bar, which is the target material in this paper. Ceramizable composites can be coated on the surface of a copper conductor and sintered to form a ceramic insulating layer, so that the bus bar can work in some specific environments. The application for the ceramic insulation of a metal conductor requires not only a good processability of the composites, but also a lower sintering temperature and a higher strength for the ceramic. Most conductors in power equipment are copper, which melts at 1083 °C, so the ceramic should be sintered at a low temperature of no more than 1000 °C. In order to get ceramics with better properties, the temperature program must be designed and controlled accurately.

There are many achievements in the research on the formulation and processing of ceramizable silicone rubber composites [11,12,13,14,15,16,17,18,19,20,21,22,23], but research on the ceramization process and mechanism is not enough. Hanu et al. [1,2,3] researched the ceramization mechanism of mica/silicone rubber composites, and put forward that ceramization was a process where the edge of mica melted and reacted with silica, which was the residue of silicone rubber, and the liquid phase connected the fillers and improved the residual strength. However, the temperature was too high when the reaction between mica and silica occurred, and no low-melting-point fluxing agent was used in these studies, so the temperature of the ceramization was over 1000 °C while the maximum flexural strength was only 8 MPa. Mansouri et al. [4] improved the low-temperature ceramization by adding glass frit into silicone rubber-based composites, and researched the ceramization mechanism further. It was reported that the glass frit was the key to reducing the sintering temperature, because the glass frit could melt to form a liquid phase and connect the mica and silica at a lower temperature. However, the explanation of the mechanism was still based on results at 1000 °C, and the mechanism at a lower temperature was not discussed. In recent years, a few researchers studied the ceramization process and mechanism at lower temperatures, and the flexural strength of the ceramics sintered at 1000 °C was improved to 20 MPa [24,25,26,27]. However, the ceramization performance is still poor, especially at a lower temperature, and the ceramization mechanism should be improved.

Researching the ceramization process and mechanism can provide theoretical guidance for the formulation design of ceramizable composites, help to improve ceramization properties and reduce the sintering temperature. In the early research, the strength of ceramics increased obviously only if the sintering temperature reached 1000 °C. In this research, ceramizable composites with different contents of nano silica were prepared, and the ceramic samples were sintered at different temperatures of no more than 1000 °C. The ceramization process and mechanism below 1000 °C were discussed in detail. Low-temperature sintering at 800 °C was achieved and the strength of ceramics was improved. The low-temperature ceramization performance was improved considerably. What is more, the effect of nano silica on the ceramization was investigated.

## 2. Materials and Methods 

### 2.1. Materials

Ceramizable silicone rubber composites were prepared by methyl vinyl silicone rubber, kilchoanite, low-melting-point glass frit, nano silica, hydroxyl silicone oil and 2,4-dichlorobenzoyl peroxide (DCBP). Silicone rubber had good processibility as the matrix. Kilchoanite (Ca_24_Si_16_O_56_) was the main inorganic filler for ceramization. Low-melting-point glass frit was the fluxing agent, which could melt within the range of 400–500 °C to form a liquid phase. Nano silica could improve the mechanical properties of the silicone rubber composites, participate in the ceramization reaction, and improve the strength of ceramics. Hydroxyl silicone oil could soften the silicone rubber to improve the processability of the composites. 2,4-dichlorobenzoyl peroxide (DCBP) was the vulcanizing agent of the silicone rubber. The components of the low-melting-point glass frit were analyzed by XRF and are shown in Table 1.

Table 2 presents the formulations of the ceramizable silicone rubber composites. All of the fillers and agents were added into the silicone rubber, in the order from left to right in Table 2. S0–S5 were the samples with different contents of nano silica, so as to research the effect of nano silica on the ceramization. S6 was used to research the effect of the sintering temperature on the properties of the ceramics. The content of the glass frit in S6 was increased properly based on S3, so that the density and strength of the ceramic samples could be improved [28].

### 2.2. Preparation of Ceramizable Composites and Ceramic Samples

The materials were mixed by a two-roller internal mixer at 50 °C for a better mixture of fillers and silicone rubber, and the speed of rollers was 30 rpm. If the temperature was too high, DCBP would decompose and the crosslink reaction would take place in advance. Then, the samples were molded and vulcanized in a steel mold (100 mm × 100 mm × 2 mm) at 120 °C for 10 min, and degassed in an oven at 150 °C for 4 h, so that the ceramizable silicone rubber composites were prepared.

To get the ceramic samples, ceramizable silicone rubber composites were cut into strip-shaped samples (80 mm × 10 mm × 2 mm), buried by Al_2_O_3_ powder and sintered in the furnace with the program shown in Figure 1. The temperature program was designed according to the decomposition process of the composites. The sintering temperature of S0–S5 was 1000 °C (Step 10, X °C in Figure 1). There were five different sintering temperature for S6: 600 °C, 700 °C, 800 °C, 900 °C, and 1000 °C (Step 10, X °C in Figure 1).

### 2.3. Characterization of Ceramizable Composites and Ceramic Samples

First of all, thermogravimetric analysis (TGA; TGA/SDTA851, METTLER TOLEDO, Zurich, Switzerland) was carried out in nitrogen with a heating rate of 10 °C/min. A proper temperature program for the sintering of the ceramic samples was designed according to the decomposition process of the composites, and the ceramic samples were sintered in the furnace. The linear contraction of the ceramic samples was calculated by Equation (1):*L* = (*L*_0_ − *L*_1_)/*L*_0_(1)
where *L* is linear contraction (%), *L*_0_ is original length of the samples before sintering (mm), and *L*_1_ is length of the samples after sintering (mm).

The flexural strength of the ceramic samples was measured with an electronic universal testing machine (CMT4503, MTS Industrial System, Shenzhen, China). The length of the supporting span in three-point bending test was 30 mm, and the speed of the applied load was 2 mm/min. More details of the three-point bending test are provided in the Appendix A. The microstructure of fracture surface for ceramic samples S6 sintered at different temperatures was observed with a scanning electron microscope (SEM; VE-9800S, KEYENCE, Osaka, Japan), in order to investigate the microstructure evolution during ceramization. Finally, the X-ray diffraction (XRD) analysis of the ceramic samples was carried out on a diffractometer (D8 ADVANCE A25, BRUKER, Karlsruhe, Germany). The ceramic samples were ground into powder and the XRD patterns were recorded over the 2*θ* range of 10–65°.

## 3. Results and Discussion

### 3.1. Thermogravimetric Analysis of Ceramizable Composites

Figure 2 shows the results of the thermogravimetric analysis for the ceramizable composites sample S6. The composites without glass frit and the composites without kilchoanite were prepared based on the formulation of S6, so as to investigate the decomposition process of the sample S6. Thermogravimetric analysis of these two samples and kilchoanite powder were also carried out and are shown in Figure 2. There were two mass loss stages in S6 (Figure 2a). The first one was the decomposition of the silicone rubber matrix. It was within the temperature range of 400–480 °C, which was almost the same temperature range as the melting of the glass frit (400–500 °C). Meanwhile, the first decomposition process of the sample without the glass frit mainly took place within the temperature range of 500–560 °C. It indicated that the glass frit could reduce the decomposition temperature of silicon rubber. Glass frit melted at first as the temperature rose up, and metal ions such as K^+^ and Ca^2+^ in the glass frit had a catalytic effect on the decomposition of the silicone rubber, so that the melting of the glass frit and the decomposition of the silicone rubber took place simultaneously. It was reported that the melting point of the fluxing agent should be lower than the decomposition temperature of the polymer matrix, or else the inorganic fillers would run off during the decomposition of the matrix [29].

Kilchoanite caused the second decomposition process of S6, according to Figure 2b. There was no second stage in the sample without kilchoanite, and the second stage in S6 was similar to the decomposition process of kilchoanite. Moreover, the glass frit could also accelerate the decomposition of kilchoanite (Figure 2a). It indicated that kilchoanite probably reacted with the melted glass frit during 600–700 °C. The temperature program for the sintering of ceramic samples was designed as shown in Figure 1, according to the decomposition process of the composites. The aims of Step 4 to Step 8 were to make the matrix decompose slowly, to produce enough liquid phase, and to prevent inorganic fillers from running off. The matrix of S6 decomposed completely at 500 °C. Kilchoanite in S6 decomposed mainly from 600 to 700 °C. The residual mass was stable from 710 to 800 °C.

### 3.2. Ceramization Performance at Different Temperatures

S6 was sintered at five different temperatures (600 °C, 700 °C, 800 °C, 900 °C and 1000 °C) in order to investigate the reaction between kilchoanite and glass frit, and the ceramization process and mechanism of the composites. Figure 3a–e shows the microstructure of the fracture surface for the ceramic samples sintered at different temperatures. Figure 3f shows the XRD patterns of the ceramic samples. Because kilchoanite was the only crystal in the composites, all the samples were compared with it to judge whether there was a new crystal in the ceramic samples.

The structure of the 600 °C-sintered sample was incompact (Figure 3a). The XRD pattern of 600 °C was almost the same as kilchoanite (Figure 3f), and kilchoanite particles could be observed in Figure 3a. The components did not react to form new crystals. Kilchoanite and silica were only stuck together by the melted glass frit. In the 700 °C-sintered sample, kilchoanite disappeared in both the SEM image (Figure 3b) and XRD pattern (Figure 3f). There was some continuous glassy phase and incompact silica in Figure 3b. Kilchoanite decomposed and reacted with the glass frit to form the glassy phase. The XRD pattern showed that there was no other crystal, but only silica in the 700 °C-sintered sample. When the sintering temperature reached 800 °C, there was a new crystal petedunnite (CaZnSi_2_O_6_) in the XRD pattern. The existence of Ca and Zn in petedunnite proved that both kilchoanite and glass frit participated in the reaction to form the crystal. A completely connected microstructure had formed in the sample, but there were still a lot of pores and silica (Figure 3c). When the temperature kept on rising, the silica was invisible in both the SEM images and XRD patterns. The porosity of the samples reduced and the intensity of petedunnite in the XRD patterns increased gradually (Figure 3d–f). It indicated that silica also participated in the reaction to form petedunnite.

Figure 4 shows the linear contraction and flexural strength of sample S6 sintered at different temperatures. Both the linear contraction and flexural strength increased rapidly from 600 to 800 °C, but increased slowly from 800 to 1000 °C. It could be explained perfectly by the SEM images and XRD patterns. When the sintering temperature was 800 °C, the flexural strength was 46.76 MPa. When the sintering temperature reached 1000 °C, the flexural strength was improved to 54.56 MPa. In the early research, the strength of the ceramics increased obviously, only if the sintering temperature reached 1000 °C, and the flexural strength was no more than 20 MPa [24,25,26,27]. In this research, 800 °C was the key point of the ceramization performance, which was reduced by 200 °C compared with other research.

### 3.3. Effect of Nano Silica on Ceramization

S0–S5, which had different contents of nano silica, were sintered at 1000 °C. Figure 5 shows the XRD patterns of the ceramic samples. The crystal phase of S0, which had no nano silica, consisted of mainly hardystonite (Ca_2_ZnSi_2_O_7_), some perovskite (Ca_4_Ti_4_O_12_), and zinc phosphate (Zn_2_P_2_O_7_), but no petedunnite (CaZnSi_2_O_6_). Hardystonite and petedunnite were present in sintered sample S1, which had nano silica of 10 parts per hundred of rubber (phr). Then, hardystonite disappeared in the XRD patterns of S2–S5, and the intensity of petedunnite in the XRD patterns increased gradually from S1 to S3. Moreover, there were silica and cristobalite in S4 and S5, which had excess nano silica. The intensity of the cristobalite increased greatly and the intensity of the silica also increased in S5. Because the surface of sintered samples S0 and S1 was damaged, Al_2_O_3_ could be observed in the XRD patterns of S0 and S1.

Figure 6 shows the linear contraction and flexural strength of sintered samples S0–S5. Both first increased and then decreased as the content of the nano silica increased. The maximum appeared at the point of 30 phr. Therefore, a proper content of the nano silica could improve the strength of the ceramic samples effectively. It should be noted that the content of the inorganic fillers was counted based on the mass of the silicone rubber, which was convenient for the preparation of ceramizable composites, but the silicone rubber decomposed and did not participate in the ceramization at all. It was the proportion between each of the inorganic fillers that determined the ceramization performance of the composites. If the content of kilchoanite increases to 100 phr, then the content of nano silica should increase to 60 phr. A proper proportion of kilchoanite and nano silica was 5:3 in weight. Of course, the proportion between the polymer matrix and inorganic fillers could also affect the ceramization performance, but it was not within the scope of the discussion in this part. What is more, the strength of S3 was 42.81 MPa, which was lower than the S6 sintered at 1000 °C. As mentioned before, the content of the glass frit in S6 was increased properly based on S3, so that the density and strength of the ceramic samples could be improved [28].

Nano silica played an important role in the formation of the ceramic phase in this research. In order to discuss the effect of the nano silica on ceramization clearly, the formula of hardystonite is written as CaO·0.5ZnO·SiO_2_ and the formula of petedunnite is written as CaO·ZnO·2SiO_2_. The mol ratio of Ca:Zn:Si in hardystonite was 1:0.5:1, while the mol ratio of Ca:Zn:Si in petedunnite was 1:1:2. There was more Zn and Si in the petedunnite. Ca was from the kilchoanite and Zn was from the glass frit. Si was from the kilchoanite, glass frit, and nano silica (there was no nano silica in S0). The content of kilchoanite or glass frit of samples S0–S5 was the same, but the content of nano silica was different. For sample S0, which had no nano silica, although the glass frit and kilchoanite could provide enough Si, there was no petedunnite, but only hardystonite in the sintered sample. Nano silica was necessary to form petedunnite. When enough nano silica was added into the composites, more glass frit participated in the reaction, and there was no hardystonite, but only petedunnite in the sintered sample. Then, the strength of the ceramic samples increased greatly. However, when the nano silica was in excess, only a part of nano silica could take part in the formation of petedunnite, and the rest would turn into crystalline silica and cristobalite. This made the strength of the ceramic samples obviously decrease, as there were defects in the ceramic phase. On the one hand, nano silica had a large specific surface area and high surface energy, which improved the sintering kinetic force and reaction speed; on the other hand, nano silica could migrate efficiently and make the reaction take place more evenly in samples, so that the sintering could be completed at a lower temperature. In addition, when the temperature reached 700 °C, the glass frit had melted and kilchoanite also decomposed, then nano silica acted as the frame of the material. Therefore, when the nano silica was inadequate, the material would lose the support and the liquid phase could run off, so the strength of samples S0–S2 was much lower than that of S3.

### 3.4. Ceramization Process and Mechanism

Ceramization process could be divided into two stages according to whether petedunnite formed. The first stage was below 800 °C. The glass frit melted to form a liquid phase, and accelerated the decomposition of silicone rubber matrix. The liquid phase enfolded the kilchoanite and silica, which finished at about 500 °C. During 600–700 °C, kilchoanite decomposed and reacted with the melted glass frit to form the glassy phase, and the nano silica acted as the frame of the material instead of kilchoanite. Then, as the temperature rose up, the glassy phases connected with each other to form a stable microstructure. Although the ceramic phase did not form, the sample had a certain mechanical strength because of the glassy phase. The second stage began at a temperature of no more than 800 °C. The glassy phase reacted with the silica, petedunnite appeared and the ceramic phase formed. The material had a completely connected microstructure with the ceramic phase and glassy phase bonded together, so that both the contraction and strength increased obviously. Finally, as the temperature rose up to 1000 °C, more silica participated in the reaction and the crystal petedunnite grew more perfectly. The density and strength of the samples increased gradually. 

In conclusion, ceramization mechanism of ceramizable silicone rubber composites is liquid-phase sintering. The ceramic samples had a high strength and low sintering temperature, but the contraction problem was serious [30,31]. There are two keys of ceramization at a low temperature, the proper content of nano silica and low-melting-point glass frit. Glass frit melts at a low temperature to form a liquid phase, which is beneficial to substance transfer, such as flow, diffusion, dissolution, and precipitation. Nano silica plays the role of a frame to support the material, and then takes part in the ceramization reaction, reducing the sintering temperature and improving the strength of the ceramic samples. Figure 7 shows the ceramization process and mechanism of ceramizable silicone rubber composites. Small pores formed at 500 °C because the silicone rubber decomposed, but the liquid phase could fill up most of the pores. Then, kilchoanite decomposed and reacted with the liquid phase, and the reaction caused contraction as well, so there were more pores of a larger size. Finally, more and more pores were filled up by the formation and growth of petedunnite, but the size of the samples contracted more obviously because of the reaction.

## 4. Conclusions

In this research, ceramizable silicone rubber composites used for the ceramic insulation of a metal conductor, such as a bus bar, were prepared. Low-temperature sintering at 800 °C was achieved by adding low-melting-point glass frit and nano silica. The low-temperature ceramization performance of the composites was improved considerably. When the sintering temperature was 800 °C, the flexural strength was 46.76 MPa. When the sintering temperature reached 1000 °C, the flexural strength improved to 54.56 MPa. The ceramization process was divided into two stages. The first stage was the formation of the glassy phase and the second stage was the formation of the ceramic phase. There were two keys to ceramization at low temperature, the proper content of nano silica and low-melting-point glass frit. The contraction of the ceramic samples was not low enough, because the proportion of silicone rubber matrix was about 47 wt.% in S6. If the proportion of the matrix was reduced, the contraction of the ceramic samples would decrease and the flexural strength could be improved further.

## Figures and Tables

**Figure 1 materials-13-03708-f001:**
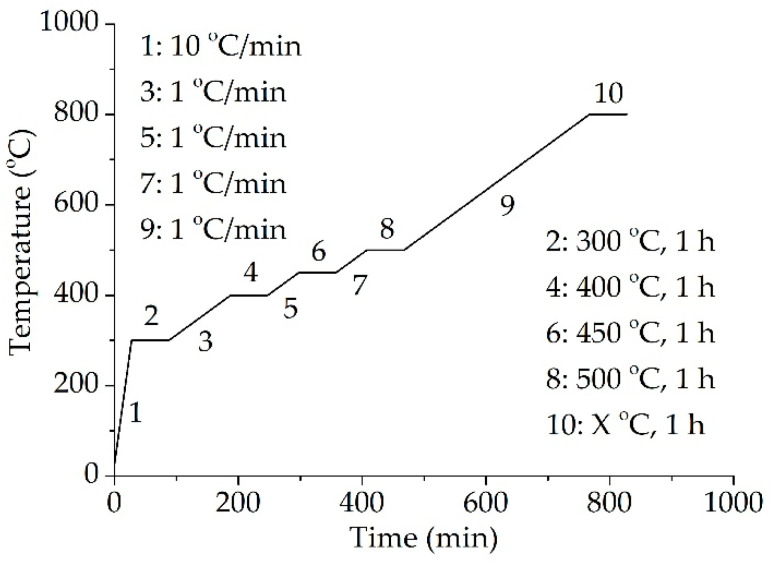
Temperature program for the sintering of the ceramic samples.

**Figure 2 materials-13-03708-f002:**
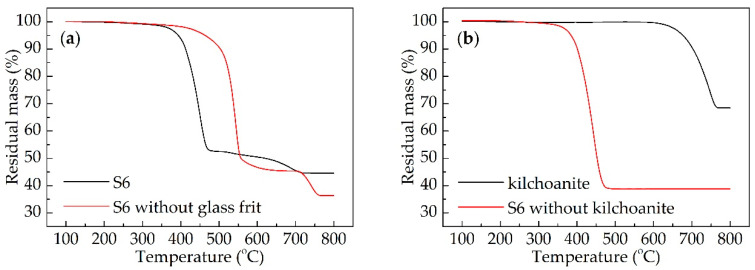
Thermogravimetric curves of different composites samples: (**a**) S6 and S6 without glass frit; (**b**) kilchoanite powder and S6 without kilchoanite.

**Figure 3 materials-13-03708-f003:**
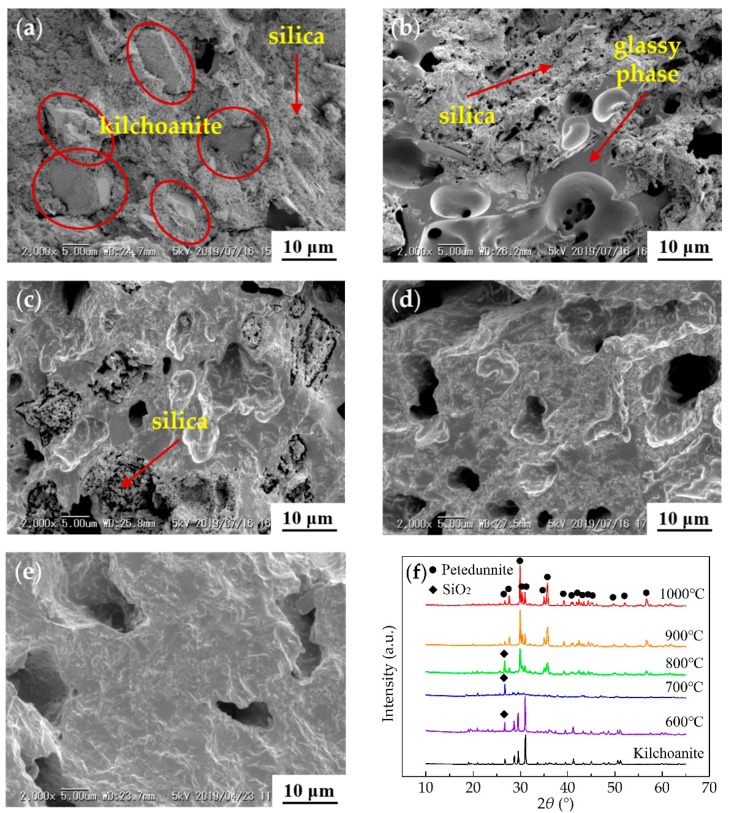
SEM images of samples sintered at different temperatures: (**a**) 600 °C; (**b**) 700 °C; (**c**) 800 °C; (**d**) 900 °C; (**e**) 1000 °C; and (**f**) XRD patterns of different samples.

**Figure 4 materials-13-03708-f004:**
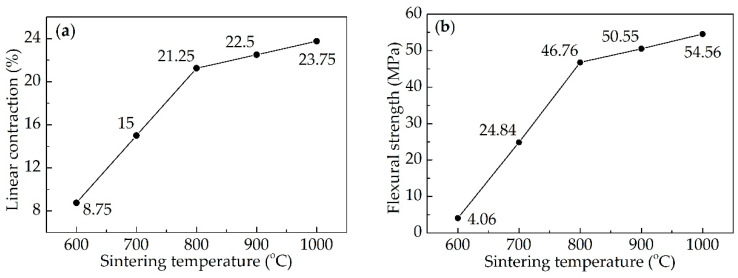
Properties of the ceramic samples sintered at different temperatures: (**a**) linear contraction; (**b**) flexural strength.

**Figure 5 materials-13-03708-f005:**
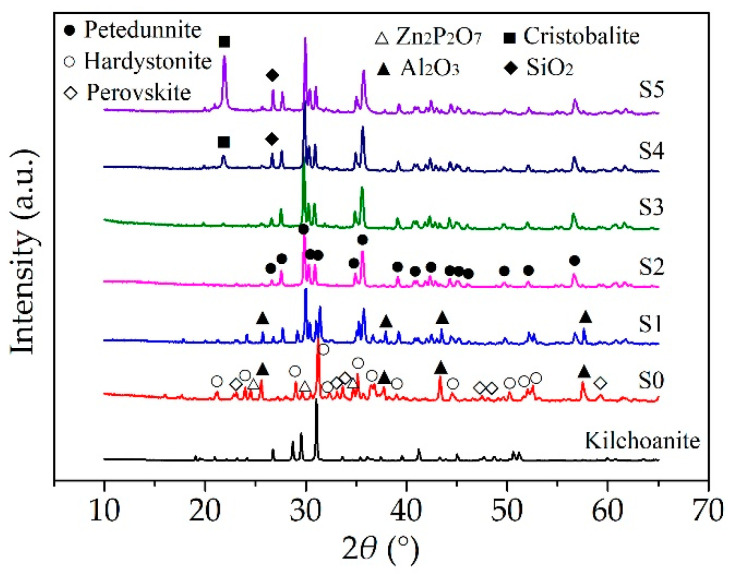
XRD patterns of the sintered samples with different contents of nano silica.

**Figure 6 materials-13-03708-f006:**
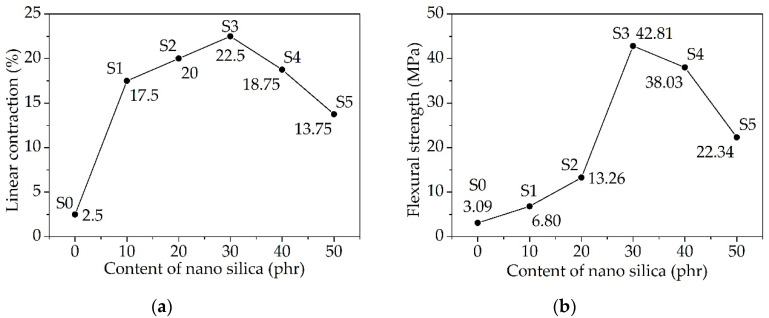
Properties of ceramic samples with different contents of nano silica: (**a**) linear contraction; (**b**) flexural strength.

**Figure 7 materials-13-03708-f007:**
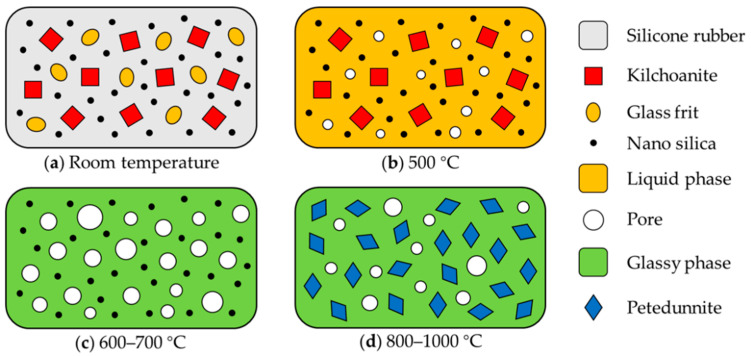
Ceramization process and mechanism of ceramizable silicone rubber composites.

**Table 1 materials-13-03708-t001:** The analyzed components of the low-melting-point glass frit.

Components	ZnO	SiO_2_	TiO_2_	P_2_O_5_	K_2_O	CaO	Others
Content (wt.%)	43.58	27.64	10.86	7.56	7.38	2.34	0.64

**Table 2 materials-13-03708-t002:** The formulations of the ceramizable silicone rubber composites (phr ^1^).

Samples	Silicone Rubber	Hydroxyl Silicone Oil	Nano Silica	Kilchoanite	Glass Frit	DCBP
S0	100	3	0	50	25	1.5
S1	100	3	10	50	25	1.5
S2	100	3	20	50	25	1.5
S3	100	3	30	50	25	1.5
S4	100	3	40	50	25	1.5
S5	100	3	50	50	25	1.5
S6	100	3	30	50	30	1.5

^1^ parts per hundred of rubber.

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
