# Peer review of "Ceramization Mechanism of Ceramizable Silicone Rubber Composites with Nano Silica at Low Temperature"

_materials, 2020, doi:10.3390/ma13173708_

Round 1

Reviewer 1 Report

I have found your paper as interested due to using of kilchoanite as a phase which could stabilize composite microstructure changes during ceramization process. In present form the paper is a concious raport of investigation on model sintering process conducted in very low procedure. My general opinion about this experiment is good. In fact, application of ceramizable composites as a coverings of electrical cables demands also proper behaviour of such materials under fast degradation of silicone phase. In my opinion investigations should be fulfilled by rapid ceramisation investigations. The most probably, such process will cause intensive gas production which will result in porosity creation. Such effect should decrease material strength but it also should limits linear contraction. The strength is not the most important property of such materials, in my opinion more important is cohesion between ceramised material and copper wire. Anyway, my opinion is a suggestion of further investigations. In present form, the paper could be printed as a "Part 1" of larger investigation plan.

Reviewer 2 Report

The paper contains interesting work on sintered silicon rubber composite in respect to their ceramization mechanism based on temperature and composition of the composite. In general, the manuscript reads well and the science has been presented in a clear way. The only criticism would be lack of details on mechanical characterisation, which should be improved.

Authors very briefly introduced the 3 point bending test in chapter 2.3 providing the length of the sample and the speed of applied load. This must be elaborated a little and should contain more data, such as full geometry of the specimen (length, width, depth, notch/no notch) and also fracture load. Authors mention that the fracture surface has been investigated by SEM, but no results are seen, please include. The load-displacement curves would be also useful to see by potential readers and since the cermitized rubber composites become ceramic Young’s modulus and fracture toughness values calculated from the test would significantly increase the value of the manuscript. I also suggest providing more information on the formulas used to calculate values for the benefit of the readers.  

Reviewer 3 Report

The manuscript is well written and could be accepted after addressing the following issues: 

  1. 2nd paragraph in Introduction: Author could include the general strength requirement for the materials applied for the coating on copper conductors. 
  2. Figure 1. xC for 1 hour seems confusing to me. Author may turn it into a table to present the heating profile.
  3. More details needed on the measurement of the strength, e.g. what standard (ISO, AMST).
  4. Figure 3f: alongside the morphology evolution, author needs to comment on the chemical compatibility with Cu at different heating temperatures.
  5. for Figure 7, the reaction equations may be needed to explain the phase evolution. Meanwhile, the author may comment on the formation of porosity (with bigger size?)
  6. English should be carefully editted. 
